# Impact of Bi and Sn on Microstructure and Corrosion Resistance of Zinc Coatings Obtained in Zn-AlNi Bath

**DOI:** 10.3390/ma13173788

**Published:** 2020-08-27

**Authors:** Henryk Kania, Mariola Saternus, Jan Kudláček

**Affiliations:** 1Department of Advanced Materials and Technology, Faculty of Engineering Materials, Silesian University of Technology, Krasińskiego 8, 40-019 Katowice, Poland; Henryk.Kania@polsl.pl; 2Department of Metallurgy and Recycling, Faculty of Engineering Materials, Silesian University of Technology, Krasińskiego 8, 40-019 Katowice, Poland; 3Department of Manufacturing Technology, Czech Technical University in Prague, Technická 4, 166-07 Prague, Czech Republic; Jan.Kudlacek@fs.cvut.cz

**Keywords:** corrosion resistance, hot dip coatings, galvanizing baths, corrosion tests

## Abstract

The paper presents results of studies on the impact of bismuth and tin additions to the Zn-AlNi bath on microstructure and corrosion resistance of hot dip galvanizig coatings. The structure at high magnifications on the top surface and cross-section of coatings received in the Zn-AlNiBiSn bath was revealed and the microanalysis EDS (energy dispersion spectroscopy) of chemical composition was determined. The corrosion resistance of the coatings was tested relatively in a neutral salt spray test (NSS), and tests in a humid atmosphere containing SO_2_. Electrochemical parameters of coatings corrosion were determined. It was found that Zn-AlNiBiSn coatings show lower corrosion resistance in comparison with the coatings received in the Zn-AlNi bath without Sn and Bi alloying additions. Structural research has shown the existence of precipitations of Sn-Bi alloy in the coating. It was found that Sn-Bi precipitations have more electropositive potential in relation to zinc, which promotes the formation of additional corrosion cells.

## 1. Introduction

Galvanized steel is currently one of the basic materials used in construction. Steel constructions [1] and rods used for concrete reinforcement [2] are commonly corrosion-protected by means of hot dip-galvanized coatings. For the production of fences, galvanized wires are used, which show very good pullability [3], while galvanized sheets with good pressing ability are used for roofing [4]. Zinc coatings provide barrier and sacrificial protection of steel, at the same time they show very high resistance to mechanical damage [5], which determines the constantly growing demand for this type of corrosion protection, including even high-strength alloy steels after heat treatment [6].

Hot dip galvanizing technology currently consumes more than half of global zinc production [7]. At the same time, this technology generates the largest losses of zinc as a result of its inefficient use for coating formation. This is due to the production of zinc-rich waste such as zinc ashes and hard zinc, but also zinc losses may result from insufficient zinc dripping from the top surface of product when removing it from the bath and the production of coatings with excessive thickness. This means that the direction of development of the modern process of hot dip galvanizing is the drive to reduce the consumption of zinc and its effective use to produce a coating. The above goal is achieved by appropriate selection of alloying additions for galvanizing baths. Nowadays, they are multi-component zinc baths including different alloying elements, which are added to the zinc bath simultaneously, the effect of which will give [8,9]: contraction of the bath surface oxidation intensity, restraining the impact of silicon content in steel on coating extension, developing the ability of liquid Zn to drip from a top surface of the product while being brought out from a bath.

Excessive oxidation of the bath surface leads to the generation of a large amount of waste in the form of zinc ashes. An effective way to reduce it is to add aluminium to a bath. Already at a content of 0.005 wt.% Al forms an alumina barrier film on the bath surface, which protects it from further oxidation [10,11]. However, Al reacts favorably with the flux, hence its content in the bath is limited to 0.01 wt.% [5].

The reduction of the reactivity of the steel due to the Si content is achieved by introducing Ni into the bath. To achieve its effective impact, the Ni content in the bath should not be less than 0.04 wt.% [12,13]; however, exceeding the content of 0.06 wt.% increases the formation of hard zinc [14].

Betterment of the ability to flow of liquid zinc from the top surface of a product during ascent from a bath is provided by interchangeably used lead, bismuth and tin additions. Lead reduces liquid zinc surface tension and neatens the fluidity of the bath [15,16]; on the other hand due to its toxicity and harmful impacts on both human health and natural environment, its application as an alloying element to Zn bath is restrained and is increasingly being removed from the bath [17,18]. Non-toxic bismuth becomes an option to the addition of lead; what is more, the application of 0.1 wt.% Bi provides a similar intensity of liquid zinc flowing from the surface as about 1 wt.% Pb [19], which means that its required content in the bath is almost 10 times smaller. Bismuth can also advantageously diminish the solubility of iron in the liquid zinc [20]. Studies have shown, however, that bismuth presence may promote the phenomenon of liquid metal embrittlement (LME) [21]. Therefore, it is recommended to limit its content to the total content of Pb + 10 Bi below 1.5 wt.% [22]. Often, the supplement for bismuth, while eliminating Pb is the addition of Sn, which also improves the zinc fluidity. It is recommended that the Sn content in the bath should not exceed 0.1 wt.% [22] since this metal is also considered a factor increasing the tendency to crack steel as a result of the LME phenomenon [23]. However, EN ISO 1461 [24] allows for an entire content of alloying additions up to 1.5 wt.% but excluding iron and tin. This encourages manufacturers to increase the tin content, which is currently used most often together with bismuth in an amount of up to 0.3 wt.%.

The main purpose of using alloying bath additions is to improve the efficiency of zinc application and minimize its consumption. When choosing the bath composition, the problem of the effect of such alloying elements on coatings corrosion resistance is virtually completely ignored. The article presents the structural aspects of reducing the corrosion resistance of coatings produced in the bath with the optimal configuration of aluminium, nickel, tin and bismuth additions, applied for batch hot dip galvanizing of steel structural elements.

## 2. Experimental

### 2.1. Materials

Tests were conducted to evaluate the impact of the combined addition of Bi and Sn for hot dip galvanizing bath on the coatings microstructure and corrosion resistance. The studied bath also contained standard aluminium and nickel additions and was designated Zn-AlNiBiSn. As a comparative bath, a bath containing aluminium and nickel additions of comparable content without the addition of bismuth and tin was used, which was designated as Zn-AlNi. The Al addition in the form of ZnAl4 mortar, the Ni addition in the form of ZnNi0.5 mortar and Bi 99.99% and Sn 99.99% additions in the form of pure metals were introduced into the zinc bath prepared from super high grade (SHG) zinc 99.995%. The concentration of alloying additions was within the range that is considered optimal. Furthermore, Fe was dissolved in the bath to reach saturation state. The chemical composition selected in this way ensured the optimal interaction of alloying elements and stability of the composition during tests. Table 1 shows the composition of the examined baths, which was determined with the ARL 3460 emission spectrometer (Thermo ARL, Waltham, MA, USA).

Coatings for tests were produced on a laboratory dip galvanizing stand (Remix S.A., Świebodzin, Poland) equipped with a 3.2 dm^3^ SiC crucible. The samples were immersed in the bath for 180 s, the temperature of which was 450 °C. Before immersion in the bath, the samples were subjected to acid degreasing by immersion in HydronetBase (SOPRIN S.r.l., Maserada Sul Piave, Italy) solution (5 min), digestion in 12% hydrochloric acid solution (Chempur, Piekary Śląskie, Poland) (10 min), rinsing in water and fluxing in TegoFlux60 (Dipl. Ing. Herwig GmbH, Hagen, Germany) solution for 2 min and drying at 120 °C for 15 min. After removal, the samples with dimensions of 50 × 100 × 2 mm were air-cooled. The coatings were made on S235JRG2 low-silicon steel (Arcerlor Mittal, Dąbrowa Górnicza, Poland) containing 0.138 wt.% C, 0.021 wt.% Si, 0.743 wt.% Mn, 0.0086 wt.% S, 0.0088 wt.% [25].

### 2.2. Research Scope and Methodology

The scope of research included direct comparative corrosion tests and potentiodynamic test. Comparative tests of coatings obtained in Zn-AlNi and Zn-AlNiBiSn baths included neutral salt spray (NSS) test and test in humid atmosphere containing SO_2_. Corrosion tests were supplemented with the characteristics of coating microstructure. This scope of research allowed for simulating long-term corrosion properties of coatings and determining electrochemical corrosion parameters. Corrosion resistance tests combined with the results of microstructure tests enabled effective assessment of the corrosion resistance of coatings.

#### 2.2.1. Neutral Salt Spray (NSS) Test

The NSS test was conducted keeping with EN ISO 9227 [26]. The CORROTHERM Model 610 salt chamber made in Erichsen (Hemer, Germany) was used for the tests. In a 400 dm^3^ chamber, mist was sprayed with 5% NaCl (Chempur, Piekary Śląskie, Poland) in distilled H_2_O, whose pH was in the range 6.8–7.2. The temperature was kept at 35 °C during the tests.

The smoothness and changes on the surface of the samples were monitored every 24 h. Gravimetric tests without removing corrosion products from the surface of the samples were performed after 24, 48, 96, 240, 480, 720, 1000 h of research in salt chamber. The concluding result was the average of 5 samples of the same type and 3 measurements for each sample.

#### 2.2.2. Corrosion Test in Humid Atmosphere Containing SO_2_

Corrosion test in humid atmosphere containing SO_2_ was conducted according to EN ISO 6988 [27]. The Hygrotherm model 519 Koesternich chamber made in Erichsen (Hemer, Germany) was used for the research, which was conducted in daily cycles. One daily cycle covered 8 h of sample exposure in a closed chamber in the temperature 40 ± 2 °C and 16 h of exposure in the ambient atmosphere. For one test cycle, 2 dm^3^ of distilled water was poured into a chamber with a working volume of 300 dm^3^ and 0.2 dm^3^ SO_2_ was dosed. The smoothness and changes on the surface and gravimetric tests of the samples were performed after each daily cycle. The final result was the average of 5 samples of the same type and three measurements for each sample.

#### 2.2.3. Potentiodynamic Test

Potentiodynamic tests were performed to determine the electrochemical parameters of the corrosion process of coatings received in Zn-AlNi and ZnAlNiBiSn baths.

The Potentiostat/Galvanostat PG201 device made in Radiometer (Copenhagen, Denmark) was used for the potentiodynamic tests. The Voltamaster 1 software (Radiometer, Copenhagen, Denmark) was used to record the polarization curves. A normal calomel electrode (Radiometer, Copenhagen, Denmark) was used in the tests, which is used as a standard reference electrode. To standardize the test results, the potential values were transformed to the normal hydrogen electrode (NHE) moving the measured values by 244 mV. A calomel electrode was placed in an intermediate vessel connected to the Ługin capillary. The use of Ługin capillary prevents the influence of diffusion processes on the current and voltage read. In order to increase the accuracy of indications, the calomel electrode was immersed in the electrolyte through an electrolytic bridge filled with saturated KCl. A platinum electrode (Radiometer, Copenhagen, Denmark)was used as an auxiliary electrode.

Samples for testing after ultrasonic degreasing in trichloroethylene (Chempur, Piekary Śląskie, Poalnd), for 180 s, rinsing in distilled water and air drying were placed in a holder providing an electrode active surface of 1.0 cm^2^. In order to prevent the adsorption of pollutants during electrolysis, the electrolytic vessel was first cleaned with a highly oxidizing mixture of concentrated H_2_SO_4_ and 30% H_2_O_2_ solution (Chempur, Piekary Śląskie, Poalnd) in a ratio of 2:1, and then rinsed repeatedly with distilled water. Potentiodynamic studies were carried out in the 3.5% solution of NaCl in distilled water in 20 °C.

The open cell potential measurement lasted approx. 5 min. After determining the electrode potential in the examined environment, the dependence of current density on potential was recorded at a scan speed of 0.5 mV/s. These curves were the basis for determining the potential (E_corr_) and current density (j_corr_) of corrosion. The electrochemical parameters of the corrosion were specified by extrapolation of Tafel curves.

#### 2.2.4. Microstructure Characterization

Corrosion tests were preceded by a preliminary assessment of the coating structure. The coatings structure before the corrosion tests was disclosed using a GX51 light microscope made by Olympus (Tokyo, Japan). The image was used with analySIS software Olysia m3 (Olympus Corporation, Tokyo, Japan)). The average thickness of the coatings was determined on the basis of thickness measurements in 10 randomly selected places on 3 tested samples from the same bath. Local thickness tests were performed using an Elcometer 355 magnetic induction meter (Manchester, England).

Tests of the microstructure of coatings at high magnifications were performed using a Hitachi S-3400 N scanning electron microscope (SEM) (Tokyo, Japan); which was equipped with energy dispersion spectroscopy (EDS), on which microanalysis of chemical composition in characteristic structural components on the top surface and cross-section of the coating received in the Zn-AlNiBiSn was specified. For the registration of microstructure (SEM) and chemical composition in micro-regions (EDS) software from Noran Instruments—System Six (Thermo Fisher Scientific, Waltham, MA, USA) was applied.

## 3. Results

### 3.1. Top Surface Outlook, Cross-Section and Thickness of Coating

Immediately after manufacture, the coating received in the Zn-AlNiBiSn bath had a shiny smoothness, which asserted the attendance on the surface of the top layer of the zinc bath alloy (Figure 1a). The coating showed no lumps or discontinuities. On the comparative surface of the Zn-AlNi coating, especially in its middle part (Figure 1b), the fine-crystalline structure of the top layer of the coating is seen. This may be due to insufficient bath flow from the surface and its accumulation in this zone of the sample.

Figure 2 shows the structure of the coatings produced on the research samples before corrosion tests. Coatings received in the Zn-AlNiBiSn and Zn-AlNi bath, do not show significant divergences in the structure. The look of the coating cross-section points out the inherence of a transition zone consisted of intermetallic phases of the Fe-Zn system and the η top layer. The transition zone from the ground side build from an intermetallic phase layer δ_1_ (with a uniform thickness and compact structure), which is overlaid by the ζ intermetallic phase layer. The construction of the intermetallic phase layer ζ is more heterogeneous. Especially in the coating received in the Zn-AlNi bath, unevenness in the thickness of the phase layer ζ can be noticed (Figure 2b) This is most likely due to small fluctuations of the Si content in steel. However, the existence of Ni in the bath eliminates the influence of Si in the substrate. In the coating received in the Zn-AlNiBiSn bath (Figure 2a), such great differences in the thickness of the phase layer ζ are not observed, and its separation border with the top layer η is more even. The coating top layer (related to η—Fe solid solution in Zn) is in fact the alloy layer of the galvanizing bath. Such structure of coatings should be contemplated to be distinctive for coatings made on low-silicon steel.

The coatings thickness of the studied is shown in Figure 3. The determined average thickness after immersion time 180 s was 54.45 µm for the coating received in the Zn-AlNiBiSn bath and 52.92 µm for the coating received in the Zn-AlNi bath, respectively.

Irrespective of the bath composition, the coatings thickness is almost equal and fulfills the demands for the minimum and average value required by the EN ISO 1461 standard. The thickness of the coating produced in the bath containing Bi and Sn alloying additions is slightly higher. It can be stated that the thickness of the Fe-Zn intermetallic layers [12,13] is reduced by nickel, while bismuth and tin improve drainage of zinc bath from the surface of the product and reducing the thickness of top layer of the coating [19,25,28]. The measurement of the thickness of the Zn-AlNiBiSn coating did not show a reduction in the coating thickness compared to the thickness of the Zn-AlNi coating.

### 3.2. Top Surface Microstructure of Coating

The coating top layer (solidified Zn bath alloy layer) comes up with a shiny and bright view and nonetheless, plays protective functions in the beginning period of application the coating in corrosive conditions. Figure 4 presents the coating surface microstructure (in the bath with Al, Ni, Bi and Sn alloying elements); whereas Table 2 shows the composition in the coating characteristic micro-areas. Microanalysis of composition conducted on the coating surface from the area of 0.25 mm^2^ (Figure 4a, point 1, Table 2) indicated the occurrence of 2.4 wt.% Sn, 1.8 wt.% Bi and 95.8 wt.% Zn. The specific content of Sn and Bi on the surface of the coating is much higher than the content of these metals in the bath.

On the surface of the coating, targeted precipitations are visible, the location of which may indicate that they fill the interdendritic spaces in the zinc matrix. At high magnifications it can be seen that these precipitates have a clearly elongated (Figure 4b, point 3) or regular shape (Figure 4b, point 2). The determined average chemical composition in points two and three (Table 2) allows to state that they contain both large amounts of Sn and Bi. The morphology of the area marked by six (Figure 4c) and the high concentration in it also Sn and Bi confirms the occurrence of fine precipitations of these metals in the pure zinc matrix (point 7 wt.%–100 wt.% Zn).

As claimed by Massalski [29] and Malakhov [30] in the given Bi-Zn equilibrium system (phase diagram Bi-Zn), bismuth does not reveal solubility in solid zinc. Besides the Sn-Zn equilibrium system presented by Fries et al. [31] and Ohtani et al. [32] does not indicate tin solubility in solid state. It should therefore be taken on that Bi and Sn isolate from the solid zinc solution. The precipitates formed on the surface of the coating contain both Sn and Bi. At the same time, the lack of separate Sn and Bi precipitations suggests that these metals are released in the form of a Sn-Bi alloy in the zinc matrix.

The construction of the zinc-based precipitates is more complex. In the precipitation microstructure (Figure 4c), two phases can be observed: dark areas, constituting the matrix (point five), containing 66.2 wt.% Sn and 9.4 wt.% Bi and bright areas of lamellar shape (point four), containing 80.3 wt.% Bi and 3.6 wt.% Sn. The two-phase structure of the precipitates can be explained by the Bi-Sn equilibrium system. As claimed by Okamoto [33] on basis of the Bi-Sn equilibrium system, the solubility of Bi in Sn in solid state decreases from ~ 10.1 wt.% at eutectic temperature of 139 °C to 1.7 wt.% at room temperature forming a solid solution β (Sn).

The solubility of tin in bismuth at eutectic temperature is ~ 1.4 wt.%, while tin does not dissolve Bi at ambient temperature. On the authority of Lee [34] Sn does not form a solid solution with Bi at all. In contrast, the solubility of Bi in Sn varies from ~ 21 wt.% at eutectic temperature to 2 wt.% at ambient temperature. Because the Bi content in the resulting precipitations on the coating surface exceeds the solubility in tin, bismuth separates from the solution forming lamellar, light precipitates in the darker matrix of the β (Sn) solution.

### 3.3. Cross-Sectional Microstructure of Coating

The microstructure of the coating received in the Zn-AlNiBiSn bath together with the marked areas of EDS X-ray microanalysis is presented in Figure 5; whereas Table 3 shows the percentages of the studied elements. The microanalysis of the chemical composition made it possible to state that Bi and Sn form precipitations on the surface of the top layer, but also in the diffusion layer of the coating. However, no precipitates were found in the cross section of the top layer of the coating. 

The performed chemical composition tests in point nine showed a content of 100 wt.% zinc. Al and Ni were not found in the top layer or their content was below the limit of detection by EDS. However, microstructure tests on the cross-section of the coating confirmed the occurrence of precipitates on the surface of the coating (Figure 5a, point eight) containing both Sn and Bi (58.3 wt.% Sn, 33.1 wt.% Bi). This may indicate that the Sn and Bi, or rather the Sn-Bi alloy, is pushed out of the zinc solution during crystallization of the top layer.

The melting point of Bi is 271.4 °C, while Sn—231.9 °C. Bi with Sn form a simple equilibrium system with eutectics at 139 °C [33,34]. Assuming that the precipitates formed are a Sn-Bi alloy, their melting point is much lower than the melting point of bismuth, tin as well as zinc (419.5 °C) [26,27,28,29]. Thus, during the crystallization of the top zinc layer, the precipitations are still in a liquid state until the zinc layer solidifies completely. According to Kopyciński [35], during the crystallization of the top layer at the interface the ζ phase/liquid zinc, the heterogeneous nucleation of zinc crystals begins. The emerging embryos are characterized by dendritic growth. This effect occurs despite the lower temperature on the surface of the still liquid top layer. In areas between dendrites, unrestricted heat flow towards the surface is still possible, and supercooling in these areas is higher, which promotes the growth of zinc crystals. On this basis, it can be assumed that the crystallization front of the top layer shifts away from the surface of the diffusion layer, which due to the lack of solubility in the solid state will promote the push of Sn and Bi into the liquid phase towards the surface and its final separation from the solution on the surface of the coating after the crystallization process of the top layer.

In the diffusion layer of the coating (Figure 5c), the specific chemical composition is in agreement with the content of components in the intermetallic phases of the Fe-Zn system: in points 10 and 11—phase ζ (5.9 and 6.2 wt.% Fe), in point 12—phase δ_1_ (9.4 wt.% Fe) and at point 13—phase Γ (23.5 wt.% Fe). Nickel and aluminium contents were not found in these phases or their concentration is negligible, below EDS detection. In the area of the diffusion layer, no Bi and Sn content were found, which may indicate a lack of solubility of these metals in the intermetallic phases of the Fe-Zn system. However, precipitations of the Sn-Bi alloy were observed at the ζ phase grain boundaries (Figure 5d). Point 14 specifies the content of 33.4 wt.% Sn and 12.8 wt.% Bi. Due to the unlimited solubility of Sn and Bi in liquid zinc [29,30,31,32], it should be assumed that the precipitates were formed during the ζ phase increase. To explain this phenomenon, it can be assumed that Bi and Sn do not dissolve in Fe-Zn phases, although currently no data are available in the literature. As a consequence of this, the increase in Fe-Zn intermetallic phases leads to the precipitation of inclusions of a separate phase being the Sn-Bi alloy and pushing these particles out into the liquid zinc, which shows unlimited solubility of these metals in the liquid state. Considering that the shaping of the ζ phase layer structure occurs as a result of simultaneous processes of diffusive growth, dissolution and secondary crystallization [36], it can be assumed that most Bi and Sn pass from the zinc solution, but some remain “trapped” in the spaces between the grain boundaries of ζ phase by secondary crystallizing particles of this phase. When analyzing the location of Sn-Bi precipitates in the ζ phase layer, it was indeed possible to observe their greater concentration in the upper zone of the ζ phase, resulting from dissolution and secondary crystallization (marked in point 10), and on the border of separation with the zone of compact structure, resulting from diffusive growth (marked in point 11). However, Bi and Sn precipitations were not observed in the δ_1_ phase layer, which is characterized by a compact structure. It may therefore suggest that the presence of Sn and Bi precipitates is possible in the heterogeneities of its structure, and their excess will be pushed into liquid zinc. However, the above hypothesis cannot be unambiguously accepted until experimental data on the solubility of Bi and Sn in the intermetallic phases of the Fe-Zn system are available.

Using the assumption of the lack of solubility of Bi and Sn in Fe-Zn intermetallic phases, Pankert [37] claims that the Sn-Bi phase is separated at the ζ/top layer boundary forming a continuous layer. Considering the good solubility of Bi and Sn in liquid zinc, it is unlikely that the Sn-Bi phase layer will stick here at the time of immersion in the zinc bath. This phase must separate from the zinc solution when the top layer of the coating solidifies. Pistofidis et al. [38] showed that in baths containing Bi without the addition of Sn, Bi precipitations in the top layer of the coating are formed. Avettand-Fènoël et al. [39], on the other hand, claim that Sn precipitations are formed in the bath containing Sn both within the top layer and on its surface. Pankert [37], although the bath contained both Sn and Bi, did not find separate precipitation of these metals in the top layer. He showed, however, that Sn and Bi are pushed out of the top layer in the form of a Sn-Bi alloy on the border with the ζ phase layer. The tests carried out in the Zn-AlNiBiSn bath indicate a similar mechanism of Bi and Sn ejection in the form of a Sn-Bi alloy on the border of the top layer. However, in this case Bi and Sn transport was carried out in the opposite direction to the surface of the coating.

### 3.4. Results of Corrosion Resistance

#### 3.4.1. NSS Test

The evaluation of the corrosion resistance of the coating received in the Zn-AlNiBiSn bath was based on the appearance of corrosion changes on the surface after exposure to the salt chamber and on the basis of unitary mass changes during the test in relation to the coating received in the Zn-AlNi bath without bismuth and tin as alloying additions. View of the coating surface after 1000 h of research (Figure 6a) indicates a significant share of red corrosion products on the surface identified as iron corrosion products [40]. The Zn-AlNi coating has a lower proportion of red corrosion products and they have a less intense color (Figure 6b). The greater share of white corrosion products was identified as zinc corrosion products—[41] specifies that the corrosion process of this coating is less advanced and still occurs largely in the top layer of the η phase. The greater share of red corrosion products on the top surface testifies to the ongoing corrosion of the intermetallic phases of the Fe-Zn system [42] in the diffusion layer of the coating. In addition, the surface of the Zn-AlNiBiSn coating bath revealed point penetration of the coating into the substrate (marked in red in Figure 6).

A greater proportion of red corrosion products and local breakthrough of the coating to the substrate at a very similar thickness of the coatings (Figure 3) suggests that the corrosion of the coating received in the bath containing Bi and Sn proceeded at a much higher speed.

The change in the unit mass of the Zn-AlNiBiSn and Zn-AlNi coatings during exposure in the NSS test is shown in Figure 7. During the corrosion test, the coating showed a mass increase, which after 1000 h of exposure reached 170.38 g/m^2^. The comparative Zn-AlNi coating showed a unitary weight gain of 96.09 g/m^2^ during this time. This gives about 77% more weight gain of the coating received in the bath containing Bi and Sn as alloying additions tested.

Tests [25] of corrosion resistance in NSS test (EN ISO 9227 [26]) carried out simultaneously in the same real time of alternative coatings received in the Zn-AlNiPb bath (0.0048% Al, 0.049% Ni, 0.48% Pb) showed after 1000 h of exposure increase of 157.42 g/m^2^. Taking into account that the content of Al and Ni in the tested Zn-AlNiBiSn bath (Table 1) was comparable with the content of these additions in the Zn-AlNiPb bath and comparing the Pb content with the total content of Bi + Sn (0.067% Bi + 0.28% Sn = 0.347%) it can be stated that more than 30% higher Pb content in the bath resulted in a smaller increase in the mass of corrosion products. This shows that the combined addition of Bi and Sn for the zinc bath has a greater impact on reducing corrosion resistance than the addition of lead.

#### 3.4.2. Test in Humid Atmosphere Containing SO_2_

Figure 8 presents the surface outlook of the Zn-AlNiBiSn coating and the comparative Zn-AlNi coating after the corrosion test in a Koesternich chamber. The outlook of the coatings is gray and matte after 30 test cycles and does not reveal penetration into the substrate. On the top surface of the coating received in the bath containing Bi and Sn a large spangle can be seen (Figure 8a), which is typical for baths containing such alloying additions [37,39]. However, on the coating surface in the bath without the addition of Bi and Sn (Zn-AlNi), fine zinc grains in the form of dark and bright areas were shown (Figure 8b). This proves that in both the Zn-AlNiBiSn coating and in the Zn-AlNi coating process of corrosion took place in the top layer of η phase. However, in the area of the upper edge of the sample with the coating received in the Zn-AlNiBiSn bath, a much darker view indicates the wear of the top layer of the coating, and the corrosion process at this point begins to occur in the layer of the Fe-Zn intermetallic phases. At this point, the thickness of the top layer is the smallest due to the liquid zinc dripping down the sample. However, this effect of wear of the top layer was not observed in the sample with the coating received in the Zn-AlNi bath, which indicates its better corrosion resistance.

Unitary changes in mass of the tested coatings during exposure in Koesternich chamber is shown in Figure 9. Both coatings received in the Zn-AlNiBiSn and Zn-AlNi bath in a humid atmosphere containing SO_2_ show continuous mass loss. A larger mass loss is shown by the coating received in the Zn-AlNiBiSn bath with a tendency to increase the mass loss difference over time compared to the Zn-AlNi coating. 

After the 30 cycles exposure in Koesternich chamber, the unitary mass of the Zn-AlNiBiSn coating decreased by 27.46 g/m^2^, while the weight of the Zn-AlNi coating decreased by 15.56 g/m^2^. It can therefore be concluded that in a humid atmosphere containing SO_2_ the top layer of the coating received in the Zn-AlNiBiSn bath undergoes more intense dissolution compared to the top layer of the coating received in the Zn bath without alloying additions.

Real-time tests [25] in a humid atmosphere containing SO_2_ (EN ISO 6988 [27]) of alternative coatings received in the Zn-AlNiPb bath [25] showed a weight loss of 21.56 g/m^2^. In comparison with the coating received in the Zn-AlNi bath, this results in an increase in weight loss of approx. 38% in the bath containing Pb and approx. 76% in the bath containing Bi and Sn. Thus, the combined addition of Bi and Sn (0.347%) affects the dissolution of the coating much more intensively than the addition of Pb (0.48%).

#### 3.4.3. Potentiodynamic Test

The polarization curves of the coating received in the Zn-AlNiBiSn bath and in the Zn-AlNi bath in a solution of 3.5% NaCl are shown in Figure 10. Values of corrosion current density (j_corr_) were determined by extrapolating cathode and anode Tafel lines to corrosion potential (E_corr_) [43]. The shape of the polarization curve for the coating received in the Zn-AlNiBiSn bath shows slight differences compared to the reference coating received in the Zn-AlNi bath. However, the shift of both the cathodic and anodic branches of the polarization curve of the coating received in the Zn-AlNiBiSn bath towards higher current density values and a slight shift of the corrosion potential (E_corr_) towards more negative values can be seen. 

The corrosion potentials (E_corr_) of the coating received in the Zn-AlNiBiSn and Zn-AlNi bath (Table 4) are −781.29 and −768.37 mV (vs. NHE), respectively, which indicates that these coatings provide cathode protection of steel [44]. A slightly lower potential, however, may indicate a higher corrosion rate of the coating received in the bath containing Bi and Sn additions. The coatings received in the Zn-AlNiBiSn bath also show a higher value of corrosion current density (j_corr_). The determined values (j_corr_) are −18.84 mA/cm^2^ for the coating received in the Zn-AlNiBiSn bath and −6.24 mA/cm^2^ for the coating received in the Zn-AlNi bath, respectively (Table 4). Corrosion current density (j_corr_) characterizes mass loss according to Faraday’s law [45]. Thus, its higher value for the coating received in the Zn-AlNiBiSn bath indicates a lower corrosion resistance.

Radu et al. [46] showed lower values of corrosion current in 3% NaCl solution for coatings received in baths containing Al and Bi. However, this is explained by the Al content in the bath. Abbel Hamid et. al. [47] on the basis of potentiodynamic tests, also claims that the addition of Sn to Zn bath up to 0.5 wt.% improves the corrosion resistance of coatings. However, the polarization curves of coatings received in the bath containing Sn presented in his work [47] clearly shift towards higher values of current density and tend to shift into the range of more negative potentials.

In contrast, electrochemical studies of ZnAl alloys containing Sn [48] in a 3% NaCl solution show an increase in corrosion current density with an increase in Sn content in the alloy. The authors [48] explain the reduction in alloy corrosion resistance by the occurrence of Sn precipitations on the alloy surface.

Therefore, it can be assumed that the increase in corrosion current density (j_corr_) of the coating received in the Zn-AlNiBiSn bath may be caused by the presence of BiSn alloy precipitates on the top surface of the coating. The existence of these precipitates is of great importance for the corrosive effect because Bi as well as Sn are more cathodic in relation to Zn (E°(Bi^3+^/Bi) = 0.308 V, E°(Bi^+^/Bi) = 0.5 V, E°(Sn^2+^/Sn) = −0.1375 V, E°(Zn^2+^/Zn) = −0.7618 V; vs. standard hydrogen electrode (SHE) [49]). The electrochemical corrosion process takes place in the presence of an electrolyte between two metals with different electrode potentials. In the formed corrosion cell, Zn is an anode, dissolving while the precipitations containing Bi and Sn being a cathode are in a rather passive state [50]. Consequently, zinc is used for sacrificial protection of BiSn precipitates instead of steel substrate protection. Therefore, it can be concluded from the observations that the presence of BiSn precipitates on the surface of the coatings will accelerate corrosion processes already in the initial period of their operation.

The value of the electrode potential can also explain the poorer corrosion resistance of the coatings received in the Zn-AlNiBiSn bath in neutral salt spray and in a humid atmosphere containing SO_2_ compared to the corrosion resistance of coatings received in the bath containing Pb, in which lead precipitation was observed both on the surface and on the cross-section the top layer [25]. The potential difference between the precipitation containing positive Bi and zinc is beginning to be greater than the potential difference between the Pb precipitations (E°(Pb^2+^/Pb) = −0.1262 V vs. SHE [49]) and zinc. As a result, the action of a corrosive cell formed in the coating received in a Zn-AlNiBiSn bath is more effective compared to a corrosive cell in which lead is the cathode.

## 4. Conclusions

The impact of the bismuth and tin added to Zn-AlNi bath on the coating microstructure and corrosion resistance was investigated. The main conclusions drawn from the work are summarized below:The coatings received in the Zn-AlNiBiSn bath showed poorer corrosion resistance compared to those received in the Zn-AlNi bath without Bi and Sn as alloying additions. In the conducted corrosion tests in a neutral salt spray and in a sulfur dioxide test in a humid atmosphere, these coatings showed higher unitary mass changes in comparison with the coatings received in the Zn-AlNi bath. After the tests were completed in the salt chamber, puncture points to the substrate were found, despite having a slightly greater thickness than the coating received in the Zn-AlNi bath. Potentiodynamic tests of Zn-AlNiBiSn coatings showed more negative values of corrosion potential (E_corr_) and a higher value of corrosion current density (j_corr_).The structure of Zn-AlNiBiSn coatings showed the occurrence of typical intermetallic phases: Γ, δ_1_ and ζ and the presence of Sn-Bi alloy precipitations located on the top surface of coating and partly in the intermetallic ζ phase layer. Al and Ni were not found in the structure of the coating received in the Zn-AlNiBiSn bath. The presence of BiSn alloy precipitates may cause the formation of local corrosion cells, which causes an increase in the corrosion rate of coatings.

In summary, the Bi and Sn additions, currently used as a replacement for environmentally harmful lead, reduces the corrosion resistance of coatings.

## Figures and Tables

**Figure 1 materials-13-03788-f001:**
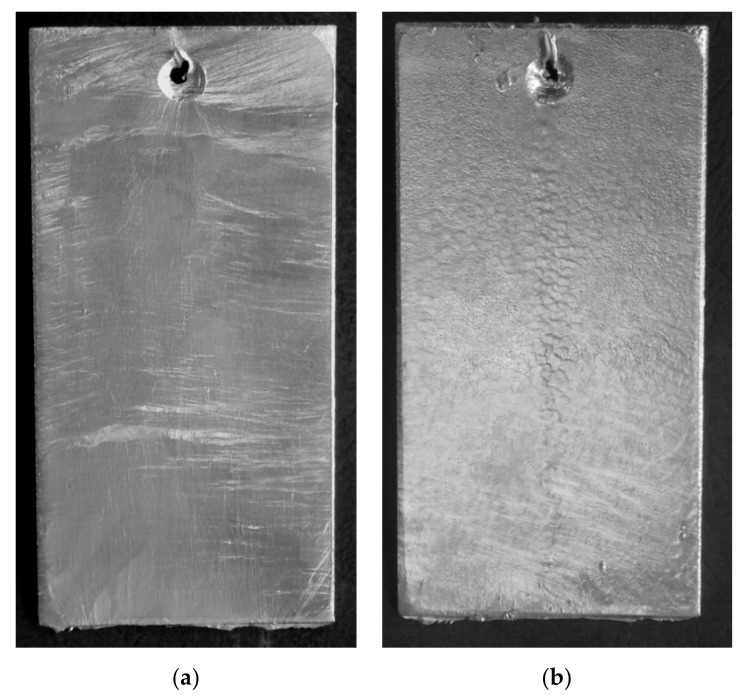
The top surface outlook of coatings made in: (**a**) Zn-AlNiBiSn, (**b**) Zn-AlNi baths.

**Figure 2 materials-13-03788-f002:**
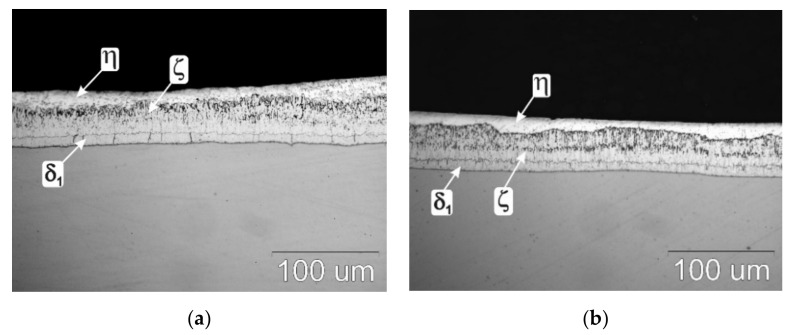
The outlook of coatings cross-section, made in: (**a**) Zn-AlNiBiSn, (**b**) Zn-AlNi baths.

**Figure 3 materials-13-03788-f003:**
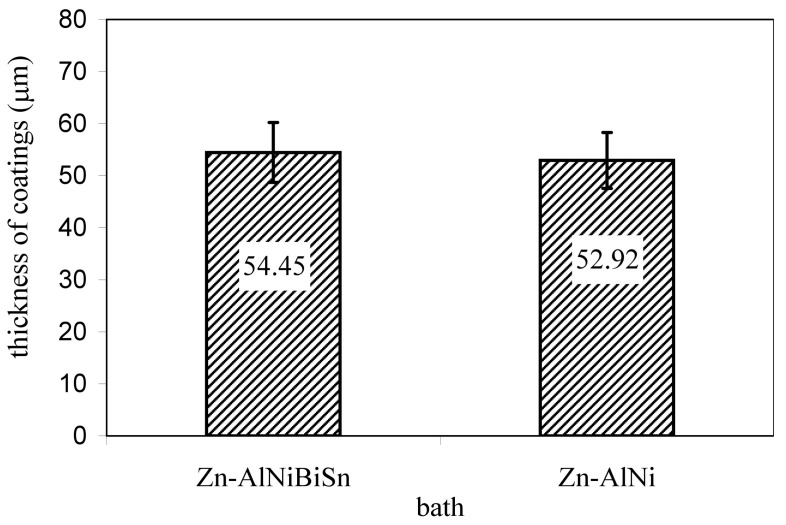
Average thickness of Zn-AlNiBiSn and Zn-AlNi coatings; immersion time 180 s.

**Figure 4 materials-13-03788-f004:**
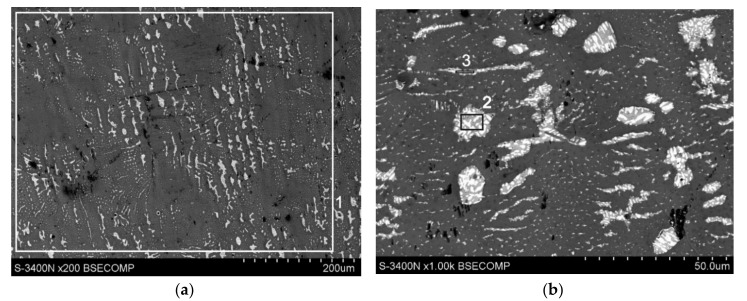
SEM images of the top surface of Zn-AlNiBiSn coating: (**a**) outlook of the coating surface, (**b**,**c**) BiSn precipitation in a zinc matrix.

**Figure 5 materials-13-03788-f005:**
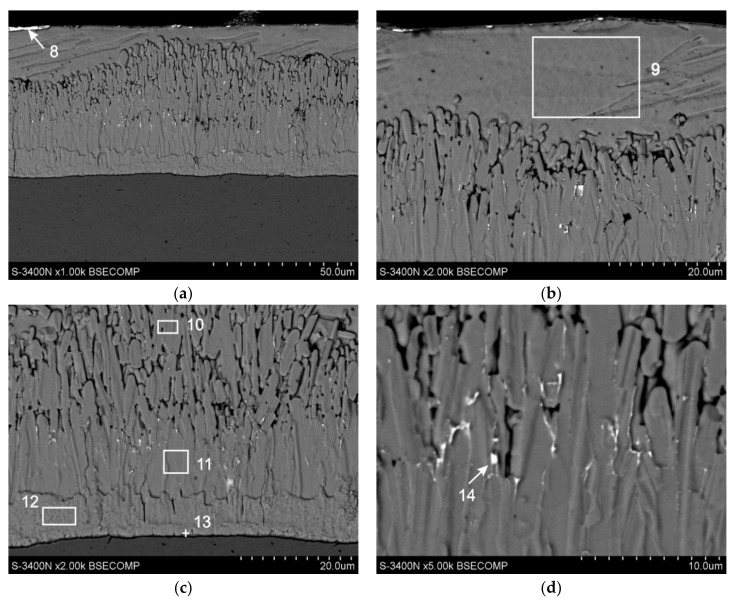
Cross-sectional microstructure (SEM) of the Zn-AlNiBiSn coating: (**a**) outlook of the coating cross-section, (**b**) top layer, (**c**) diffusion layer, (**d**) inside of ζ layer.

**Figure 6 materials-13-03788-f006:**
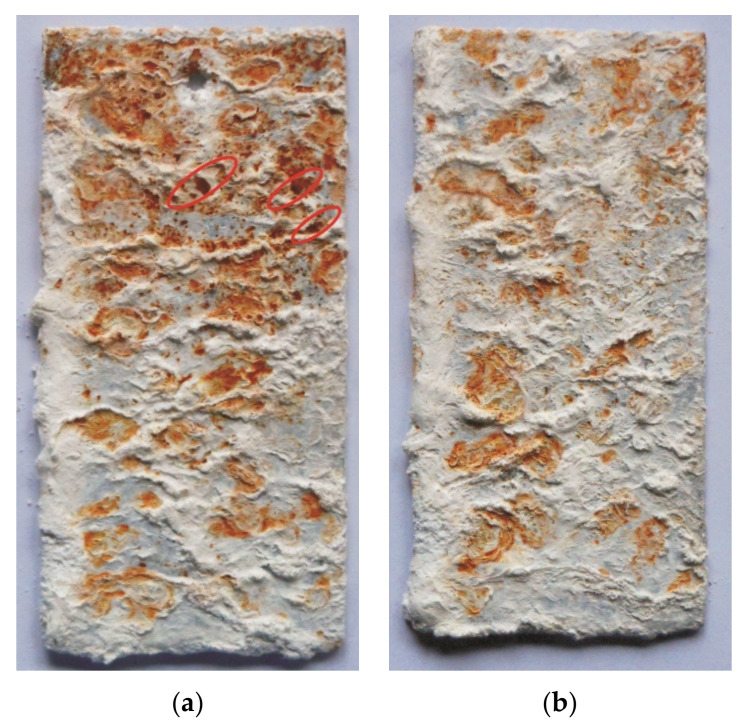
The outlook of the surface of coatings made in: (**a**) Zn-AlNiBiSn and (**b**) Zn-AlNi bath after 1000 h of research lasting in neutral salt spray (NSS) test.

**Figure 7 materials-13-03788-f007:**
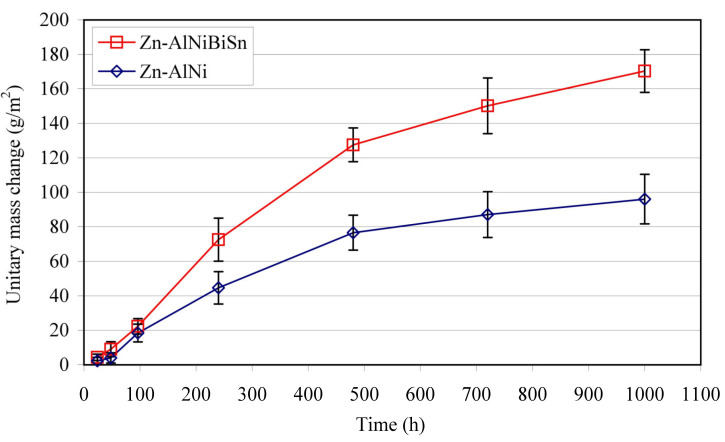
Unitary mass change of Zn-AlNiBiSn and Zn-AlNi coatings as determined from the neutral salt spray test.

**Figure 8 materials-13-03788-f008:**
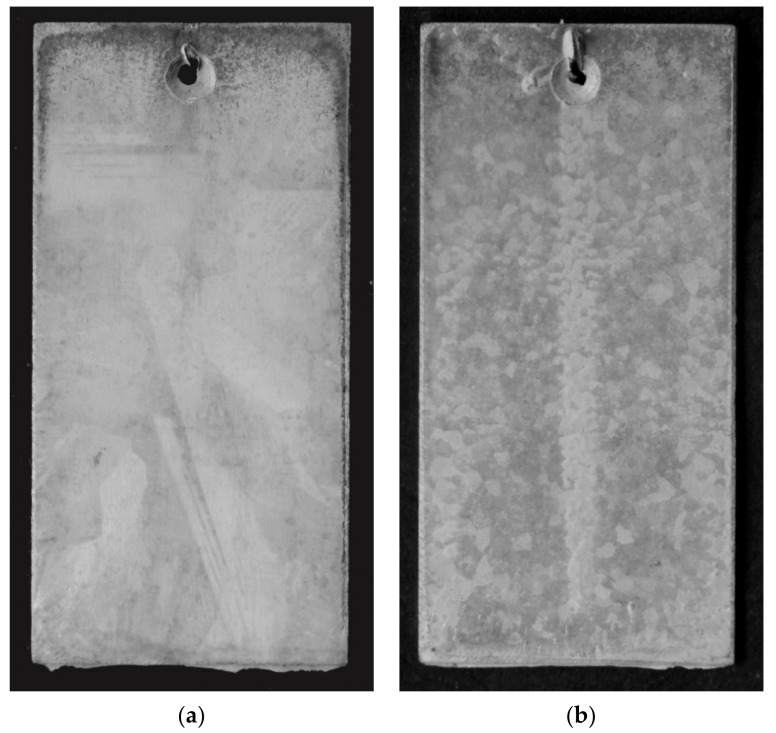
The outlook of the top surface of zinc coatings created in the: (**a**) Zn-AlNiBiSn, (**b**) Zn-AlNi baths after exposure in the Koesternich chamber.

**Figure 9 materials-13-03788-f009:**
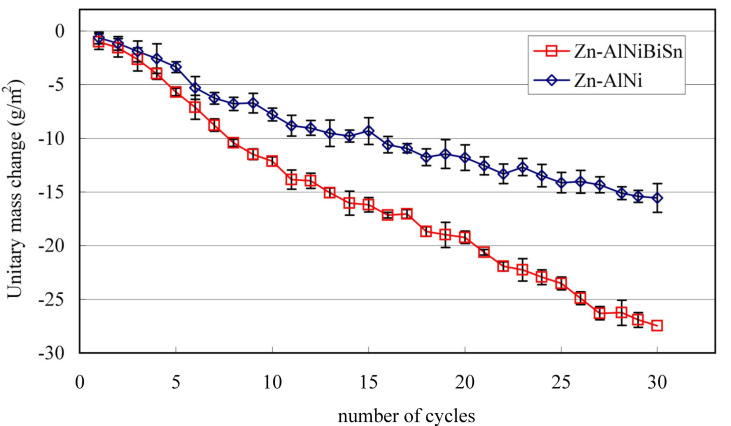
Unitary mass change of Zn-AlNiBiSn and Zn-AlNi coatings during exposure in the Koesternich chamber.

**Figure 10 materials-13-03788-f010:**
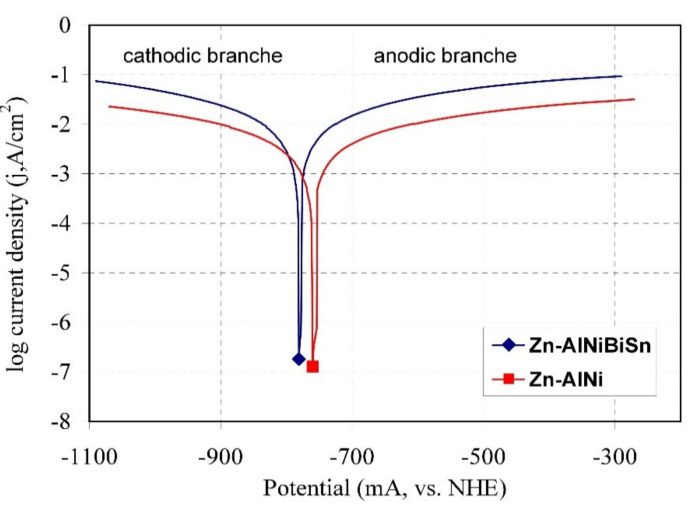
Anode and cathode polarization curves of the tested coatings in a 3.5% NaCl solution.

**Table 1 materials-13-03788-t001:** The chemical composition of the studied baths.

Bath	Content (wt.%)						
	Al	Fe	Ni	Pb	Bi	Sn	Zn and others
Zn-AlNiBiSn	0.0051	0.032	0.049	0.002	0.067	0.28	residue
Zn-AlNi	0.0054	0.034	0.052	0.002	0.0004	0.0005	residue

**Table 2 materials-13-03788-t002:** The results of the microanalysis of the composition in marked points visible on Figure 4.

Points of Microanalysis	Content of Elements					
	Zn-K		Sn-L		Bi-M	
	wt.%	at.%	wt.%	at.%	wt.%	at.%
1	95.8	98.1	2.4	1.4	1.8	0.6
2	18.1	32.0	54.2	52.7	27.7	15.3
3	53.4	70.7	31.7	23.1	14.9	6.2
4	16.1	37.3	3.6	4.6	80.3	58.1
5	24.4	38.2	66.2	57.2	9.4	4.6
6	97.3	98.8	1.6	0.9	1.1	0.3
7	100	100	-	-	-	-

**Table 3 materials-13-03788-t003:** The results of the microanalysis of the composition in marked points visible on Figure 5.

Points of Microanalysis	Content of Elements							
	Zn-K		Fe-K		Sn-L		Bi-M	
	wt.%	at.%	wt.%	at.%	wt.%	at.%	wt.%	at.%
8	8.6	16.8	-	-	58.3	62.9	33.1	20.3
9	100	100	-	-	-	-	-	-
10	94.1	93.2	5.9	6.8	-	-	-	-
11	93.8	92.8	6.2	7.2	-	-	-	-
12	90.6	89.2	9.4	10.8	-	-	-	-
13	76.5	73.5	23.5	26.5	-	-	-	-
14	49.6	64.5	4.2	6.4	33.4	23.9	12.8	5.2

**Table 4 materials-13-03788-t004:** Electrochemical corrosion parameters of the tested coatings in a 3.5% NaCl solution.

Type of Coating	Ecorr (mV, vs. NHE)	jcorr (mA/cm^2^)
Zn-AlNi	−768.37	−6.24
Zn-AlNiBiSn	−781.29	−18.84

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
