# Peer review of "Impact of Bi and Sn on Microstructure and Corrosion Resistance of Zinc Coatings Obtained in Zn-AlNi Bath"

_materials, 2020, doi:10.3390/ma13173788_

Round 1

Reviewer 1 Report

Authors reported Synergistic Effect of Bi and Sn on the Microstructure and Corrosion Resistance of Zinc Coatings Obtained in Zn-AlNi Bath. The experiment was carried out faithfully and the results was proposed. I would like to recommend a major revision before this article is published.

1. Introduction part needs to be revisited. Addition of Bi and Sn in Zn-AlNi Bath lead to lower the corrosion resistance of the coatings. Can the author explain why to add Bi and Sn? Or say, after adding Bi and Sn, what performance of the coatings can be improved.

2. Experimental: authors should give the conditions of the Potentiodynamic tests such as scanning rate, auxiliary electrode….

3. Figure 7 and figure 9. Y-axis should be Weight loss?

Author Response

Dear Reviewer,

We are grateful for taking your time to read our paper and for their constructive comments. We have carefully reviewed the comments and have revised the manuscript accordingly. Our responses are below given in a point-by-point manner. Changes to the text are shown in red in the revised manuscript. We hope the revised version is now suitable for publication.

  1. Introduction part needs to be revisited. Addition of Bi and Sn in Zn-AlNi Bath lead to lower the corrosion resistance of the coatings. Can the author explain why to add Bi and Sn? Or say, after adding Bi and Sn, what performance of the coatings can be improved.

Currently, the hot-dip galvanizing process is carried out only in zinc alloy baths, in which three groups of alloying additives are used:

 - Al to reduce surface oxidation in zinc bath,

- Ni to reduce the reactivity of steel

and

- Pb, Bi and Sn to improve the fluidity of the zinc bath.

Al and Ni are now always added to the zinc bath, hence the Zn-AlNi bath was adopted as the reference bath (the effect of these additives was described in Line 48-56). Pb, Bi and Sn are also constant alloy bath additives but are used interchangeably. Pb is currently still used at an optimal concentration of 0.4-0.5%, but due to its toxicity it is increasingly being replaced by bismuth. Bismuth, in turn, can cause the LME phenomenon, therefore its content is limited to 0.1%. In order to increase the fluidity of the bath, the Bi effect is enhanced by the addition of tin. The effect of these alloying elements is detailed in Line 57-74.

The studied configuration of AlNiBiSn alloy additives is relatively often used in industrial practice, mainly in order to improve the manufacturability of the galvanizing process and reduce the reactivity of steel in liquid zinc. Despite the benefits of using these alloying additives, the research ignores their influence on the corrosion resistance of coatings and their influence in this respect is not well understood. Therefore, the research presented in the article was undertaken.

  1. Experimental: authors should give the conditions of the Potentiodynamic tests such as scanning rate, auxiliary electrode….

The research data suggested by the reviewer were supplemented in lines 149-150 and 159-160.

A platinum electrode was used as an auxiliary electrode.

After determining the electrode potential in the examined environment, the dependence of current on potential was recorded at a scan speed of 0.5 mV/s.

  1. Figure 7 and figure 9. Y-axis should be Weight loss?

According to the reviewer's comment, the descroption "Weight loss" under Figures 7 and 9 is incorrect.

Since in the NSS test an increase in weight was observed (Fig. 7), while in the Koesternich chamber there was a weight loss (Fig. 9), the Y axis descriptions were standardized as "mass change".

Therefore, following the reviewer's remark in the captions Fig. 7 and 9, "Weight loss" was corrected to "Unitary mass change"

Yours sincerely,

Authors

Reviewer 2 Report

The topic of the paper is of practical interest and experimental research is well conducted. Results may support somehow the correlation between the corrosion resistance and presence of Sn-Bi precipitates. However, this conclusion is not corroborated with direct structural investigations, and therefore I suggest that microstructural investigations that were performed on initial layers should be extended on the depositions after corrosion tests to determine possible qualitative and/or qualitative differences regarding the corrosion mechanism. That could document one of the final conclusions i.e. “The presence of Sn-Bi precipitates may cause the formation of local corrosion cells, which causes a decrease in the corrosion resistance of coatings”.

Some valuable information could be obtained by using XRD in both initial and corroded layers, which could be corroborated for identification of phases with the EDS compositions.

Author Response

Dear Reviewer,

We are grateful for taking your time to read our paper and for their constructive comments. We have carefully reviewed the comments and have revised the manuscript accordingly. Our responses are below given in a point-by-point manner. Changes to the text are shown in red in the revised manuscript. We hope the revised version is now suitable for publication.

The topic of the paper is of practical interest and experimental research is well conducted. Results may support somehow the correlation between the corrosion resistance and presence of Sn-Bi precipitates. However, this conclusion is not corroborated with direct structural investigations, and therefore I suggest that microstructural investigations that were performed on initial layers should be extended on the depositions after corrosion tests to determine possible qualitative and/or qualitative differences regarding the corrosion mechanism. That could document one of the final conclusions i.e. “The presence of Sn-Bi precipitates may cause the formation of local corrosion cells, which causes a decrease in the corrosion resistance of coatings”.

We share the comment of the reviewer that tests of corrosion products provide very valuable information on the course of the corrosion process in a given corrosive environment. However, the XRD tests of corrosion products formed in the salt chamber on the coating obtained in the Zn-AlNiBiSn bath and the coating obtained in the Zn-AlNi bath did not show any differences in the phase composition. Taking into account that the precipitation of the BiSn alloy constitute a strongly cathode element in the structure, it should be assumed that it did not dissolve, forming corrosion products containing these metals. Moreover, the detection of small amounts of both Bi and Sn-containing corrosion products, as well as BiSn precipitations in a much larger volume of zinc corrosion products and intermetallic phases of the Fe-Zn system was not possible using the XRD method. Therefore, XRD tests of corrosion products do not add new information.

It should also be emphasized that the only variable in the process of galvanizing the test samples was the content of Bi and Sn in the bath. The content of Al and Ni in both baths was very similar and kept at the level considered optimal. The galvanizing process was also carried out with the same parameters. In the structural aspect, as well as analyzing the electrochemical properties of the system, the occurrence of BiSn alloy precipitates can therefore explain the reduction in the corrosion resistance of the coatings.

Increasing the corrosion resistance is the main purpose of using zinc coatings. The BiSn additive, which is a substitute for the toxic Pb, is relatively often used nowadays in industrial practice, mainly to improve the manufacturability of the galvanizing process and reduce the reactivity of steel in liquid zinc. Despite the benefits of using these alloying additives, the research ignores their influence on the corrosion resistance of coatings and their influence in this respect is not well understood. Therefore, the research presented in the article was undertaken, and the planned scope of the research allowed to determine their impact on corrosion resistance. However, the authors share the reviewer's suggestion regarding the influence of BiSn precipitates on the course of the corrosion process itself, therefore they conduct further research on the corrosion mechanism of zinc coatings involving BiSn precipitates, which will be the subject of subsequent publications.

Some valuable information could be obtained by using XRD in both initial and corroded layers, which could be corroborated for identification of phases with the EDS compositions.

The authors share the opinion of the reviewer that XRD tests allow for more unambiguous identification of the phase composition of coatings, especially in relation to the chemical composition in micro-areas. However, the morphology of the phases of the Fe-Zn system occurring in the zinc coating is relatively well recognized in the literature and described in the article with reference to the literature, which additionally complements the specific chemical composition in micro-areas (EDS), which is in great agreement with the homogeneity range of these phases in the system Fe-Zn.

Yours sincerely,

Authors

Round 2

Reviewer 1 Report

Authors have revised their manuscript according to reviewers' comments. I would like to recommend a minor revison before this article is published.

Comments: 

"Synergistic" is mentioned in the title of this article, how does this article reflect this feature? How to effect of Bi and Sn alone, respectively. If is not clear, the title "Synergistic Effect of Bi and Sn on the Microstructure and Corrosion Resistance of Zinc Coatings Obtained in Zn-AlNi Bath" should be changed.

Author Response

Dear Reviewer,

We are grateful for taking your time to read our paper and for their constructive comments. We have carefully rethought the comments and have revised the manuscript accordingly. 

"Synergistic" is mentioned in the title of this article, how does this article reflect this feature? How to effect of Bi and Sn alone, respectively. If is not clear, the title "Synergistic Effect of Bi and Sn on the Microstructure and Corrosion Resistance of Zinc Coatings Obtained in Zn-AlNi Bath" should be changed.

We think that the reviewer's remark about the lack of clarity of the synergy effect is correct. Taking into account the content of the article, the word "Synergistic" in the title was not fully thought out by us.

Therefore, in accordance with the Reviewer's remark, we removed the word "Synergistic" and changed the title of the article:

“Effect of Bi and Sn on the microstructure and corrosion resistance of zinc coatings obtained in Zn-AlNi bath”.

We hope the revised version is now suitable for publication.

Yours sincerely,

Authors

Reviewer 2 Report

Additions and supplementary explanations provided by the authors are satisfactory.

Author Response

Dear Reviewer,

We are grateful for taking your time to read our paper and for their constructive comments.

Yours sincerely,

Authors